# VISUAL QUESTION ANSWERING FROM ANOTHER PERSPECTIVE: CLEVR MENTAL ROTATION TESTS

## ABSTRACT

Different types of *mental rotation tests* have been used extensively in psychology to understand human visual reasoning and perception. Understanding what an object or visual scene would look like from another viewpoint is a challenging problem that is made even harder if it must be performed from a single image. 3D computer vision has a long history of examining related problems. However, often what one is most interested in is the answer to a relatively simple question posed in another visual frame of reference – as opposed to creating a full 3D reconstruction. Mental rotations tests can also manifest as consequential questions in the real world such as: does the pedestrian that I see, see the car that I am driving? We explore a controlled setting whereby questions are posed about the properties of a scene if the scene were observed from another viewpoint. To do this we have created a new version of the CLEVR VQA problem setup and dataset that we call CLEVR Mental Rotation Tests or CLEVR-MRT, where the goal is to answer questions about the original CLEVR viewpoint given a single image obtained from a different viewpoint of the same scene. Using CLEVR Mental Rotation Tests we examine standard state of the art methods, show how they fall short, then explore novel neural architectures that involve inferring representations encoded as feature volumes describing a scene. Our new methods use rigid transformations of feature volumes conditioned on the viewpoint camera. We examine the efficacy of different model variants through performing a rigorous ablation study. Furthermore, we examine the use of contrastive learning to infer a volumetric encoder in a self-supervised manner and find that this approach yields the best results of our study using CLEVR-MRT.

## 1 INTRODUCTION

Psychologists have employed *mental rotation* tests for decades (Shepard & Metzler, 1971) as a powerful tool for devising how the human mind interprets and (internally) manipulates three dimensional representations of the world. Instead of using these test to probe the human capacity for mental 3D manipulation, we are interested here in: a) understanding the ability of modern deep neural architectures to perform mental rotation tasks, and b) building architectures better suited to 3D inference and understanding.

Recent applications of concepts from 3D graphics to deep learning, and vice versa, have led to promising results. We are similarly interested in leveraging models of 3D image formation from the graphics and vision communities to augment neural network architectures with inductive biases that improve their ability to reason about the real world. Here we measure the effectiveness of adding such biases, confirming their ability to improve the performance of neural models on mental rotation tasks. Concepts from *inverse graphics* can be used to guide the construction of neural architectures designed to perform tasks related to the reverse of the traditional image synthesis processes: namely, taking 2D image input and inferring 3D information about the scene. For instance, 3D reconstruction in computer vision (Furukawa & Hernández, 2015) can be realized with neural-based approaches that output voxel (Wu et al., 2016; Nguyen-Phuoc et al., 2019), mesh (Wang et al., 2018), or point cloud (Qi et al., 2017) representations of the underlying 3D scene geometry. Such inverse graphics methods range from fully-differentiable graphics pipelines (Kato et al., 2018) to implicit neural-based approaches with learnable modules designed to mimic the structure of certain components of the forward graphics pipeline (Yao et al., 2018; Thies et al., 2019). While inverse rendering is potentially

an interesting and useful goal in itself, many computer vision systems could benefit from neural architectures that demonstrate good performance for more targeted mental rotation tasks.

In our work here we are interested in exploring neural "mental rotation" by adapting a well known standard benchmark for visual question-and-answering (VQA) through answering questions with respect to another viewpoint. We use the the Compositional Language and Elementary Visual Reasoning (CLEVR) Diagnostic Dataset (Johnson et al., 2017) as the starting point for our work. While we focus on this well known benchmark, many analogous questions of practical interest exist. For example, given the camera viewpoint of a (blind) person crossing the road, can we infer if each of the drivers of the cars at an intersection can see this blind person crossing the street? As humans, we are endowed with the ability to reason about scenes and imagine them from different viewpoints, even if we have only seen them from one perspective. As noted by others, it therefore seems intuitive that we should encourage the same capabilities in deep neural networks (Harley et al., 2019). In order to answer such questions effectively, some sort of representation encoding 3D information seems necessary to permit inferences to be drawn due to a change in the orientation and position of the viewpoint camera. However, humans clearly do not have access to error signals obtained through re-rendering scenes, but are able to perform such tasks. To explore these problems in a controlled setting, we adapt the original CLEVR setup in which a VQA model is trained to answer different types of questions about a scene consisting of various types and colours of objects. While images from this dataset are generated through the rendering of randomly generated 3D scenes, the three-dimensional structure of the scene is never fully exploited because the viewpoint camera never changes. We call our problem formulation and data set *CLEVR-MRT*, as it is a new *Mental Rotation Test* version of the CLEVR problem setup. In CLEVR-MRT alternative views of a scene are rendered and used as the input to a perception pipeline that must then answer a question that was posed with respect to another (the original CLEVR) viewpoint.[1] This gives rise to a more difficult task where the VQA model must learn how to map from its current viewpoint to the viewpoint that is required to answer the question. This can be seen in Figure 1(b). Figure 1(a) depicts a real world situation and *analogy* where the answers to similar types of questions may help different types of systems make consequential decisions, e.g. intelligent intersections, cars, robots, or navigation assistants for the blind. The fact that MRTs are a classical tool of Psychology and the link to these different practical applications motivated us to create the controlled setting of CLEVR-MRTs depicted in Figure 1(b).

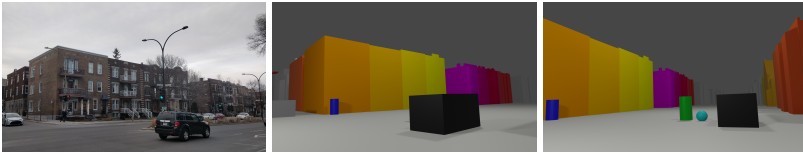

(a) (Left) A view of a street corner. (Middle) a CLEVR-like representation of the scene with abstractions of buildings, cars and pedestrians. (Right) The same virtual scene from another viewpoint, where questions concerning the relative positions of objects after a mental rotation could be of significant practical interest.

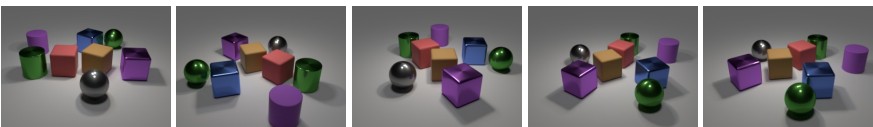

(b) Random views of an example scene in *CLEVR-MRT*. The center image is the 'canonical' view, which is the unseen point of view for which questions must be answered using only one of the other views as input.

Figure 1: (a) A real-world example where the ability to perform mental rotations can be of practical utility. (b) Images from the *CLEVR-MRT* dataset.

Using our new mental rotation task definition and our CLEVR-MRT dataset, we examine a number of new inverse-graphics inspired neural architectures. We examine models that use the FILM (Feature-wise Linear Modulation) technique (Perez et al., 2017) for VQA, which delivers competitive performance using contemporary state-of-the-art convolutional network techniques. We observe that such methods fall short for this more challenging MRT VQA setting. This motivates us to create new architectures that involve inferring a latent *feature volume* that we subject to rigid 3D transformations (rotations and translations), in a manner that has been examined in 3D generative modelling techniques such as spatial transformers (Jaderberg et al., 2015) as well as HoloGAN

---

[1]Dataset and code will be available at `https://github.com/anonymouscat2434/clevr-mrt`

(Nguyen-Phuoc et al., 2019). This can either be done through the adaptation of a pre-trained 2D encoder network, i.e. an ImageNet-based feature extractor as in Section 2.2.1, or through training our encoder proposed here, which is obtained through the use of contrastive learning as in Section 2.2.3. In the case of the latter model, we leverage the InfoNCE loss (Oord et al., 2018) to minimise the distance between different views of the *same* scene in metric space, and conversely the opposite for views of *different* scenes altogether. However, rather than simply using a stochastic (2D) data augmentation policy to create positive pairs (e.g. random crops, resizes, and pixel perturbations), we leverage the fact that we have access to many views of each scene *at training time* and that this can be seen as a data augmentation policy operating in 3D. This in turn can be leveraged to learn an encoder that can map 2D views to a 3D latent space without assuming any extra guidance such as camera extrinsics.

## 2 METHODS

We begin here by describing simple and strong baseline methods as well as upper bound estimates used to evaluate the performance of different techniques on this dataset. We then present our new approach to learning 3D features and two different ways to address this task.

### 2.1 FILM BASELINES

The architecture we use is very similar to that proposed by FILM (Perez et al., 2017), in which a pre-trained ResNet-101 classifier on ImageNet extracts features from the input images which are then fed to a succession of FILM-modulated residual blocks using the hidden state output from the GRU. As a sanity check – to ensure our models are adequately parameterised – the simplest baseline to run is one where images in the dataset are filtered to only contain *canonical views*. In this setting, we would expect the highest validation performance since the viewpoint given is the same as the canonical viewpoint (the viewpoint wrt which the question must be answered). The second and third baselines to run are ones where we use the *full dataset*, with and without conditioning on the viewpoint camera via FILM, respectively. This is illustrated in Figure 2, where we can see the viewpoint camera also being embedded before being concatenated to the question embedding and passed through the subsequent FILM blocks. Note that in the case of the canonical baseline, the viewpoint and canonical view is the same thing, so no camera conditioning is necessary.

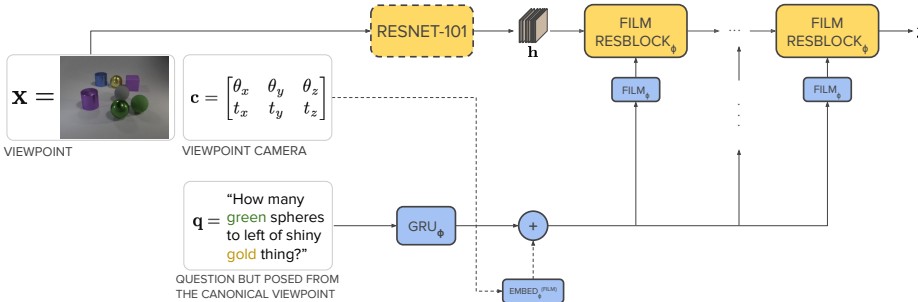

Figure 2: The pipeline of our FILM baselines (Perez et al., 2017). An input view (which is randomly sampled from a scene) is fed through a pre-trained (on ImageNet) ResNet-101 model to produce a high-dimensional stack of feature maps. The dotted border on the ResNet-101 indicate it is *frozen* during training. The question embedding is fed through a GRU which outputs an embedding vector of the sentence. The viewpoint camera is run through its own embedding $\mathbf{embed}_\phi^{\mathbf{(film)}}(\mathbf{c})$ before being concatenated to this vector (the dotted line indicates that this stage is optional, depending on what baseline is run). The resulting feature maps are fed through FILM-modulated residual blocks using the final embedding vector. (Please see the supplementary materials section for more details.)

If we let $S$ denote a scene consisting of all of its camera views (images) $\mathbf{X}$, the camera $\mathbf{c}$, the question $\mathbf{q}$, and its corresponding answer $\mathbf{y}$, we can summarise this as the following:

$$
\begin{aligned}
S = (\mathbf{X}, \mathbf{q}, \mathbf{c}, \mathbf{y}) &\sim \mathcal{D} \text{ (sample a scene)} \\
\mathbf{x} &\sim \mathbf{X} \text{ (sample a random view)} \\
\mathbf{h} &:= \text{ResNet}(\mathbf{x}) \\
\mathbf{e}_{\text{cam}} &:= \text{embed}_\phi^{(\text{film})}(\mathbf{c})
\end{aligned}
\qquad
\begin{aligned}
\mathbf{e}_{\text{gru}} &:= \text{GRU}_\phi(\mathbf{q}) \\
\tilde{\mathbf{y}} &:= \text{FILM}_\phi(\mathbf{h}, [\mathbf{e}_{\text{gru}}, \mathbf{e}_{\text{cam}}]) \\
\ell_{\text{cls}} &:= \ell(\mathbf{y}, \tilde{\mathbf{y}}), \quad\quad\quad (1)
\end{aligned}
$$

where the encoder (a ResNet) is frozen and we do not update its parameters during training.

## 2.2 LEARNING 3D FEATURE REPRESENTATIONS FROM SINGLE VIEW IMAGES

So far we have been operating in 2D, based on the pre-trained ResNet-101 ImageNet encoder which outputs a high-dimensional stack of feature maps (a 3D tensor). To work in 3D, we would either need to somehow transform the existing encoding into a 4D tensor (a stack of 3D feature *cubes*) or use a completely different encoder altogether which can output a 3D volume directly. Assuming we already had such a volume, we can manipulate the volume in 3D space directly by having it undergo any rigid transformation that is necessary for the question to be answered. In Section 2.2.1 we illustrate a simple technique which simply takes the existing ImageNet encoder's features and runs it through a learnable 'post-processing' block to yield a 3D volume, and in Section 2.2.3 we propose a self-supervised contrastive approach to learn such an encoder from scratch.

### 2.2.1 PROJECTING 2D FEATURES INTO 3D FEATURES

To exploit the power of pre-trained representations, we start with a pre-trained ResNet encoder and transform its stack of feature maps through an additional set of 2D convolution blocks using the 'post-processor' shown in Figure 3, right before reshaping the 3D tensor into 4D. In other words, we learn a module which maps from a stack of feature maps $\mathbf{h}$ to a stack of feature cubes $\mathbf{h}'$. Since the post-processor is a *learnable* module through which the FILM part of the pipeline is able to backpropagate through, it can be seen as learning an appropriate set of transforms that construct 3D feature volumes $\mathbf{h}'$. Through back-propagation it learns to perform well when manipulated with camera controllable FILM operations either as is or, more interestingly, when also subjected to rigid 3D transformations as we will see shortly in Section 2.2.2.

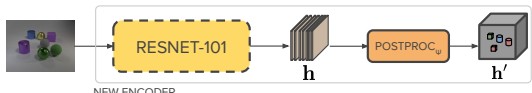

Figure 3: The pre-trained ResNet encoder outputs a stack of feature maps $\mathbf{h}$ of dimensions (1024, 14, 14), as in Fig. 2, but now a post-processing module postproc$_\psi(\mathbf{h})$, e.g. a set of 2D convolutions, processes the feature stack and reshapes it into a 4D tensor $\mathbf{h}'$ of dimensions (64, 16, 14, 14) , i.e., a stack of feature *cubes*. This entire block (inside the grey border) is the new encoder.

### 2.2.2 3D CAMERA CONTROLLABLE FILM

In lieu of conditioning the camera with FILM (as seen in Figure 2 with embed$_\phi^{(\text{film})}$), we can also condition on it to output translation and rotation parameters $(\tilde{\boldsymbol{\theta}}_x, \tilde{\boldsymbol{\theta}}_y, \tilde{\boldsymbol{\theta}}_z, \tilde{\mathbf{t}}_x, \tilde{\mathbf{t}}_y, \tilde{\mathbf{t}}_z)$ which are then used to construct a 3D rotation and translation matrix $\Theta$. Therefore, we can write out the 3D FILM pipeline as:

$$
\begin{aligned}
\mathbf{h}' &:= \text{postproc}_\psi(\text{encoder}_\phi(\mathbf{x})) \\
(\tilde{\boldsymbol{\theta}}_x, \tilde{\boldsymbol{\theta}}_y, \tilde{\boldsymbol{\theta}}_z, \tilde{\mathbf{t}}_x, \tilde{\mathbf{t}}_y, \tilde{\mathbf{t}}_z) &:= \text{embed}_\phi^{(\text{rot})}(\mathbf{c}) \\
\mathbf{h}'_{\text{rot}} &:= \text{transform}(\mathbf{h}'; P(\tilde{\boldsymbol{\theta}}_x, \tilde{\boldsymbol{\theta}}_y, \tilde{\boldsymbol{\theta}}_z, \tilde{\mathbf{t}}_x, \tilde{\mathbf{t}}_y, \tilde{\mathbf{t}}_z))
\end{aligned}
\qquad
\begin{aligned}
\tilde{\mathbf{y}} &:= \text{FILM}_\phi(\mathbf{h}'_{\text{rot}}, [\text{GRU}_\phi(\mathbf{q})]) \\
\ell_{\text{cls}} &:= \ell(\mathbf{y}, \tilde{\mathbf{y}}), \\
&\quad\quad\quad\quad\quad\quad (2)
\end{aligned}
$$

where we now minimise $\ell_{\text{cls}}$ with respect to our usual parameters $\phi$ and the post-processor parameters $\psi$, and $P(\cdot)$ is a function that produces a rigid transform matrix from its parameters (via Euler angles). This is illustrated in Figure 4. Note that in this figure we have illustrated that we can still perform camera conditioning via embed$_\phi^{(\text{film})}$, i.e. we can use the camera to do rigid transforms, modulate the

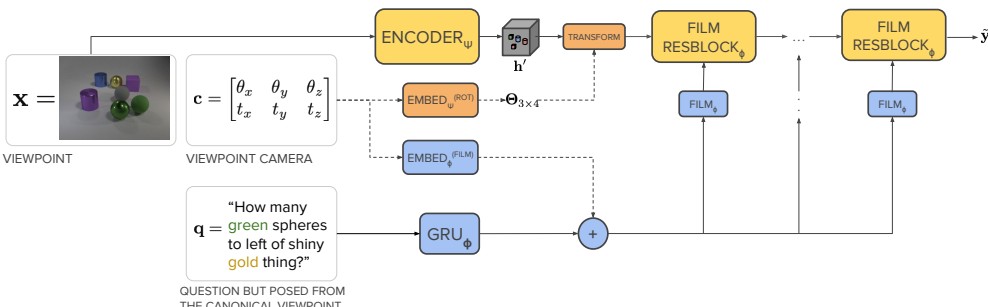

Figure 4: The 3D version of the FILM pipeline proposed in Section 2.2. The encoder can be either the 2D-to-3D formulation in Section 2.2.1 (with the ResNet-101 inside it frozen but the postprocessor block learnable, i.e. Figure 3) or the contrastive encoder in Section 2.2.3 (which has no post-processor and is completely frozen). A camera encoder $\mathbf{embed}_{\psi}^{(\mathbf{rot})}(\mathbf{c})$ is trained to map the camera coordinates of the scene to a transformation matrix which is used to transform the resulting 4D volume via an explicit rotation and translation, and/or it can be embedded and concatenated with the GRU embedding via $\mathbf{embed}_{\phi}^{(\mathbf{film})}$ (the dotted lines for both indicate that either/or are optional). This volume is then fed to FILM-modulated ResBlocks (which internally compute 3D convolutions since we are dealing with a volume) to produce the answer.

GRU embedding, or both. For the sake of brevity, Equation 2 only shows the case where we use the camera for rigid transforms via $\text{embed}_{\psi}^{(\text{rot})}$. Also note that we cannot directly use the raw camera parameters $\mathbf{c} = (\boldsymbol{\theta}_x, \boldsymbol{\theta}_y, \boldsymbol{\theta}_z, \mathbf{t}_x, \mathbf{t}_y, \mathbf{t}_z)$ to construct the rigid transform $\Theta$ is because these are relative to world coordinates. Finally, in the next section (Section 2.2.3), we will show that we can replace the encoder and its postprocessor (Figure 3) with a contrastive encoder trained from scratch, without the use of a postprocessor since it directly outputs a latent volume.

### 2.2.3 LEARNING 3D CONTRASTIVE ENCODERS

In Section 2.2.1 the encoder proposed was an adaptation of a pre-trained ImageNet classifier backbone to output latent volumes. Here we propose the training of an encoder from scratch in an unsupervised manner, via the use of contrastive learning as demonstrated in Chen et al. (2020). Conceptually it is simple: given two random views of the same scene $\mathbf{x}_1$ and $\mathbf{x}_2$, the goal of the encoder $\text{enc}_{\mathbf{h}}$ is to infer a *latent volume* from each of those views $\mathbf{h}_1$ and $\mathbf{h}_2$. To enforce the notion of similarity between these volumes (and conversely the opposite for an $\mathbf{x}_2$ that is not from the same scene as $\mathbf{x}_1$), both these volumes are reduced down to dense codes (through an extra encoder $\text{enc}_{\mathbf{z}}$) $\mathbf{z}_1$ and $\mathbf{z}_2$, where the contrastive loss is applied. This is shown in Figure 5. Unlike Chen et al. (2020) however, our main goal here is to infer *latent volumes* for downstream tasks, rather than latent codes.

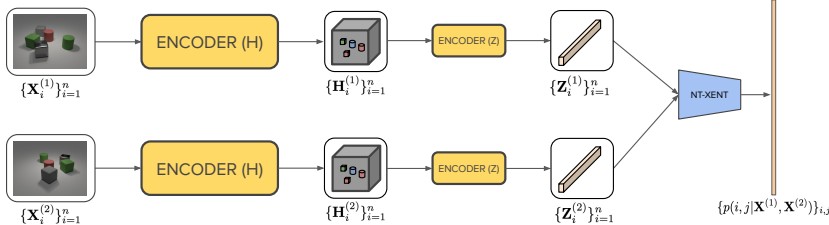

Figure 5: The contrastive-based encoder, inspired from (Chen et al., 2020). We sample two sets of minibatches $\mathbf{X}^{(1)}$ and $\mathbf{X}^{(2)}$, where the $i$'th instance in each set comprises a positive pair (different views of the same scene). The $\mathbf{H}$ encoder generates a 3D volume for each view, and an additional $\mathbf{Z}$ encoder convolves this down to a summarisation vector over which the contrastive loss is applied.

Let us denote $\mathbf{X}^{(1)}$ and $\mathbf{X}^{(2)}$ to be minibatches of images, with subscripts for individual examples in the minibatch (e.g. $\mathbf{X}_j^{(1)}$). We will assume that $(\mathbf{X}_i^{(1)}, \mathbf{X}_j^{(2)})$ correspond to the same scene if $i = j$, otherwise they are different. The InfoNCE loss (Oord et al., 2018) is defined as:

$$\mathbf{H}^{(1)} = \text{enc}_{\mathbf{h}}(T(\mathbf{X}^{(1)})), \ \mathbf{H}^{(2)} = \text{enc}_{\mathbf{h}}(T(\mathbf{X}^{(2)})), \ \mathbf{Z}^{(1)} = \text{enc}_{\mathbf{z}}(\mathbf{H}^{(1)}), \ \mathbf{Z}^{(2)} = \text{enc}_{\mathbf{z}}(\mathbf{H}^{(2)}),$$

$$\mathcal{L}_{\text{NCE}} = \frac{1}{n} \sum_{i=1}^{n} \ell_{\text{NCE}}^{(i)}, \text{ where } \ell_{\text{NCE}}^{(i)} = -\log \frac{\exp(\text{sim}(\mathbf{Z}_i^{(1)}, \mathbf{Z}_i^{(2)})/\tau)}{\sum_{k=1}^{n} \exp(\text{sim}(\mathbf{Z}_i^{(1)}, \mathbf{Z}_k^{(2)})/\tau)}, \tag{3}$$

where $T(\cdot)$ is some stochastic data augmentation operator (e.g. random crops, flips, colour perturbations) which operates on a per-example basis in the batch. This loss also contains a temperature term $\tau$, which is a hyperparameter to optimise (in practice, we found $\tau = 0.1$ to produce the lowest softmax loss). Since a large number of negative examples is needed to learn good features, we train this encoder on an 8-GPU setup with a combined batch size of 2048. Note that in Chen et al. (2020), the contrastive loss is enforced between stochastic 2D data augmentations of the same image i.e. they contrast $T(\mathbf{X}^{(1)})$ and $T(\mathbf{X}^{(2)})$ (with $\mathbf{X}^{(1)} = \mathbf{X}^{(2)}$). We refer to this as *2D only* data augmentation, since the contrastive learner does not ever contrast between two different views of a scene. In the case of Equation 3, since $\mathbf{X}^{(1)} \neq \mathbf{X}^{(2)}$ we call this *2D + 3D* data augmentation, and if $T$ is the identity function then we refer to this as *3D only* data augmentation. When training of this encoder has converged, we freeze it and use it in place of the pre-trained ImageNet encoder and postprocesor as originally shown in Figure 3, as well as the same pipeline described in Figure 4 and Equation 2.

## 3 RELATED WORK

Several extensions of the CLEVR dataset exist, though mainly through NLP-based extensions such as evaluating systematic generalisation (Bahdanau et al., 2019), adding dialogue (Kottur et al., 2019), and robust captioning of changes between scenes (Park et al., 2019). In terms of visual-based extensions, Yi et al. (2019) proposed a temporal version of CLEVR which looks at VQA in the context of causal and counterfactual reasoning. Concurrent to our work, a version of CLEVR has recently been proposed (Qiu et al., 2020) in the context of reinforcement learning, where an agent is trained to perform viewpoint *selection* on a scene to be able to answer the question, with each scene consisting of a large occluder object in the center to accentuate occlusions. However, the main difference is that our dataset decouples the camera viewpoint from the viewpoint from which the question must be answered. Furthermore, this dataset has relatively limited question and scene variability (for instance, focusing on only two types of questions and the same occluding object in the center). Lastly, in our work we do not assume the VQA model is an agent that is able to vary its active viewpoint to better answer the question – instead, our model must learn to 'imagine' what the same scene should look like from another perspective, conditioned only on a single view.

Using rigid transforms to infer latent 3D volumes was loosely inspired by HoloGAN (Nguyen-Phuoc et al., 2019). Here, they use a GAN to map a randomly sampled noise vector to a 3D latent volume before subsequently rendering using a neural renderer. Several works (Nguyen-Phuoc et al., 2018; Sitzmann et al., 2018; Lombardi et al., 2019) condition on images and camera poses to learn voxels that represent the input, with options to re-render from different camera poses. These methods, however, assume that the camera poses are known and evaluated on single-scene settings. Conversely, our dataset consists of tens of thousands of scenes, which makes any re-rendering task (i.e. autoencoding) significantly more difficult due to the need to reconstruct well on all scenes, all while having a larger computational footprint than decoder-less approaches. Neural rendering is also not limited to just voxel representations; other ways of encoding 3D data can be used such as point clouds (Qi et al., 2017), meshes (Wang et al., 2018; Kato et al., 2018), surfels (Rajeswar et al., 2020), latent codes (Eslami et al., 2018), or even in the weights of a neural network (Mildenhall et al., 2020).

Other VQA models have been proposed, e.g., MAC (Hudson & Manning, 2018) proposes a memory and attention-based reasoning architecture for more interpretable VQA. While this could in principle be modified to leverage 3D volumes, FILM serves as a simpler architectural choice for analysis. As for learning encoders, contrastive learning is currently state-of-the-art in learning self-supervised representations that are competitive with their supervised variants (Oord et al., 2018; Bachman et al., 2019; Chen et al., 2020; He et al., 2020). Tian et al. (2019) explored contrastive learning of scenes, though the multiview aspect in this setting was applied to different sensory views, rather than camera views. Harley et al. (2019) explored the use of contrastive learning on 2.5D video (i.e. RGB + depth) to predict novel views, with the goal of learning 3D object detectors in a semi-supervised manner.

The CLEVR dataset (Johnson et al., 2017) is a VQA dataset consisting of a range of synthetic 3D shapes laid out on a canvas. The dataset consists of a range of questions designed to test various aspects of visual reasoning such as counting (e.g. 'how many red cubes are in this scene?'), spatial relationships (e.g. 'what colour is the cylinder to the left of the big brown cube?') and comparisons

(e.g. 'are there an equal number of blue objects as red ones?'). In recent years however, proposed techniques have performed extraordinarily well on the dataset (Perez et al., 2017; Hudson & Manning, 2018), which has inspired us to explore VQA in more difficult contexts. The original CLEVR dataset provided one image for each scene. *CLEVR-MRT* contains 20 images generated for each scene holding a constant altitude and sampling over azimuthal angle. To ensure that the model would not have any clues as to how the view had been rotated, we replaced the asymmetrical "photo backdrop" canvas of the CLEVR dataset with a large plane and centered overhead lighting. To focus on questions with viewpoint dependent answers, we filtered the set of questions to only include those containing spatial relationships (e.g. 'is X to the right of Y'). From the original 90 question templates, only 44 contained spatial relationships. In total, the training + validation split consists of 45,600 scenes, each containing roughly 10 questions for a total of 455,549 questions. 5% of these scenes were set aside for validation. For the test set, 10,000 scenes were generated with roughly 5 questions each, for a total of 49,670 questions. Figure 1(b) shows an example of one of these scenes.

## 4 RESULTS AND ANALYSIS

For each experiment, we perform a sweep over many of the hyperparameters (which are detailed in the appendix) to find the experiment which performs the best, according to validation set accuracy. We then select the best-performing experiment and run repeats of it with varying seeds (3-6, depending on the overall variance), for a total of $N$ runs. The validation set accuracy reported is the mean over these $N$ runs, and similarly for the test set. It is worth noting that for our 3D FILM experiments, some runs appeared to hit undesirable local minima, exhibiting much lower validation accuracies, which we conjecture is due to a 'domain mismatch' between using a encoder that was initially pre-trained for ImageNet classification and our CLEVR dataset. (This appears to be supported by the fact that these outliers do not exist when we use our pre-trained contrastive encoder, in Table 2.) To deal with these outliers, we instead compute the mean/stdev over only the top three runs out of the $N = 6$ we originally trained.

Table 1: Table of results for experiments run using a pre-trained ResNet-101 encoder. The *Upper bound* model is a baseline model where *only* canonical views are given as input, and is expected to have the highest performance since it does not have to answer questions from another random viewpoint as input. For the columns shown: **3D?** refers to whether we are using 2D latents or 3D latents (the difference between Figs 2 and 4); **camera (embed)** refers to embedding the camera coordinates and concatenating it with the question embedding; **camera (rotation)** refers to using the camera to map to a rigid transform of the volume (shown in Figure 4). The result denoted with † (in small text) indicates the same experiment but with the postprocessor frozen after random weight initialisation. See Supp. 6.3 for more on MAC baseline. Accuracies shown are percentages. For all *3D FILM, projection* results, the mean/stdev is computed over the top 3 performing models (out of 6).

| Method | 3D? | camera (embed) | camera (rotation) | valid acc. (%) | test acc. (%) |
|---|---|---|---|---|---|
| Majority class | – | – | – | $24.72 \pm 0.00$ | $24.75 \pm 0.00$ |
| GRU-only | – | ✗ | ✗ | $49.38 \pm 0.40$ | – |
| Upper bound (*canon. views only*) | ✗ | ✗ | ✗ | $94.19 \pm 0.39$ | $94.24 \pm 0.40$ |
| MAC (Hudson & Manning, 2018) | ✗ | ✓ | ✗ | $70.96 \pm 0.97$ | – |
| 2D FILM (Sec 2.1, Fig 2) | ✗ | ✗ | ✗ | $70.63 \pm 0.19$ | $69.60 \pm 0.09$ |
| | ✗ | ✓ | ✗ | $83.95 \pm 1.21$ | $83.68 \pm 1.21$ |
| 3D FILM, projection (Sec 2.2.1, Fig 4) | ✓ | ✗ | ✗ | $68.19 \pm 1.87$ | – |
| | ✓ | ✓ | ✗ | $88.82 \pm 3.04$ | $86.36 \pm 3.46$ |
| | ✓ | ✗ | ✓ | $\mathbf{92.80 \pm 0.30}$ († $68.98 \pm 1.43$) | $\mathbf{90.86 \pm 0.87}$ |
| | ✓ | ✓ | ✓ | $89.83 \pm 1.36$ | $89.68 \pm 1.34$ |

Our results for the FILM baselines (Section 2.1) and using 2D-to-3D projections (Section 2.2.1) are shown in Table 1. What we find surprising is that the 2D baseline without camera conditioning is able to achieve a decent accuracy of roughly 70%. On closer inspection the misclassifications do not appear to be related to how far away the viewpoint camera is from the canonical, with misclassified

Table 2: Table of results for experiments run with the contrastive encoder described in Section 2.2.3. For the columns shown: **Data aug** refers to what data augmentation scheme was used, for stochastic data augmentation operator $T()$ ($2D$ = contrast $T(\mathbf{X}_1)$ and $T(\mathbf{X}_1)$, $3D$ = contrast $\mathbf{X}_1$ and $\mathbf{X}_2$, $2D+3D$ = contrast $T(\mathbf{X}_1)$ and $T(\mathbf{X}_2)$; **NCE accuracy** is how well the contrastive encoder is able to distinguish between pairs of views that belong to the same/different scene; the remaining columns denote the FILM task, as described in Table 1.

| Contrastive pre-training stage | | | FILM stage | | | |
|---|---|---|---|---|---|---|
| Data aug | $\tau$ | NCE accuracy (valid) | camera (embed) | camera (rotation) | valid acc. (%) | test acc. (%) |
| 2D | 0.1 | 9.13 | ✓ | ✗ | $59.78 \pm 0.23$ | $59.14 \pm 0.43$ |
| | | | ✗ | ✓ | $59.29 \pm 0.44$ | $58.57 \pm 0.53$ |
| 3D | 0.1 | 99.72 | ✓ | ✗ | $57.42 \pm 0.26$ | $56.74 \pm 0.31$ |
| | | | ✗ | ✓ | $57.63 \pm 0.21$ | $57.10 \pm 0.33$ |
| 2D + 3D | 0.1 | 98.14 | ✓ | ✗ | $65.15 \pm 4.63$ | $63.70 \pm 3.74$ |
| | | | ✗ | ✓ | $\mathbf{87.49 \pm 0.78}$ | $\mathbf{86.01 \pm 0.69}$ |

points being distributed more or less evenly around the scene. Given that the question-only baseline ('GRU-only') is able to achieve an accuracy significantly greater than that of the majority class baseline ($\approx 25\%$ versus $\approx 50\%$), it seems like it is likely exploiting statistical regularities between the actual question itself and the scene. If we add camera conditioning via FILM (that is, appending the camera embedding to the GRU's embedding) then we achieve a much greater accuracy of $83.68 \pm 1.21$. Furthermore, our results demonstrate the efficacy of using rigid transforms with 3D volumes, achieving the highest accuracy of $92.80 \pm 0.30$ on the test set. If we take the same experiment and freeze the postprocessor (denoted by the small † symbol), then we achieve a much lower accuracy of 69 %. This is to be expected, considering that any camera information that is forward-propagated will contribute gradients back to the postprocessing parameters in the backward propagation, effectively giving the postprocessor supervision in the form of camera extrinsics. (However, we will soon show just as good performance for our contrastive encoder experiments in Table 2, which did not utilise any camera extrinsics in the pre-training phase, and no postprocessor in the FILM stage.) Finally, the last row of the table shows that if one uses the camera for both rigid transforms *and* embedding, the mean test accuracy is roughly the same as the rigid-transform-only variant ($90.86 \pm 0.87$ vs $89.68 \pm 1.34$). This appears to suggest that simply performing rigid rotations of the volume is sufficient by itself for good performance.

In Table 2 we perform an ablation on the type of data augmentation used during the contrastive pre-training stage (described in Section 2.2.3) and find that 3D data augmentation is essential for the encoder to distinguish whether a pair of views come from the same scene or not, as shown in the 'NCE accuracy' column. However, both 2D and 3D data augmentation is necessary in order for the FILM task to yield the best results, as seen in the last row. Similar to Table 1, utilising the viewpoint camera for rigid transforms produces the best results, with $86.01 \pm 0.69$ % test accuracy. While the best result of Table 1 does have a *much higher* variance ($86.60 \pm 7.57$), we consider our contrastive 2D + 3D experiment to yield our best result due to its almost negligible variance.

Our results show that our best performing formulation (either 2D-to-3D or contrastive) performs on average only 8 % less than the canonical baseline's 94%, which can be seen as a rough upper bound on generalisation performance. While we obtained promising results, it may not leave a lot of room to improve on top of our methods, and we identified some ways in which the dataset could be made more difficult. One of them is removing the viewpoint camera coordinates and instead placing a sprite in the scene showing where the canonical camera is. This means that the model has an additional inference task it has to perform, which is inferring the 3D position of the canonical viewpoint from a 2D marker. Another idea is to allow some variation in the elevation of the viewpoint camera. While this can accentuate the effects of occlusion (if the camera is allowed to go lower than its default elevation), it also provides for a more grounded dataset since occlusions are commonplace in real-world datasets. We examine the latter here, generating a version of CLEVR-MRT where the camera elevation is allowed to vary, and with both small and large objects present in the scene (small objects were not present in the original CLEVR-MRT dataset). Concretely, the default elevation in the original dataset was $30°$, whereas now it is randomly sampled from a Uniform$(20, 30)$. An example scene of this dataset is shown in Figure 6.

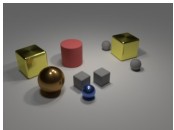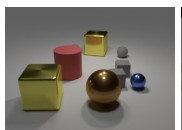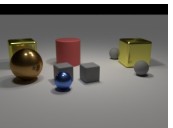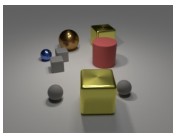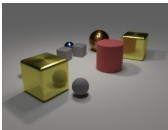

Figure 6: Left-most image is the canonical view. The rest are randomly sampled views, of varying azimuth and elevation. Compared to Figure 1b, occlusions here are more prevalent.

See Table 3 below. Please note that hyperparameter tuning for these experiments are still ongoing, with particular emphasis placed on improving the 3D FILM with camera embedding (third row) as well as the canonical upper bound (first row).

Table 3: Select experiments from Table 1 but trained on CLEVR-MRT. For all experiments shown in this table, the mean/stdev is computed over the top three runs (out of six in total).

| Method | 3D? | camera (embed) | camera (rotation) | valid acc. (%) | test acc. (%) |
|---|---|---|---|---|---|
| Upper bound (*canon. views only*) | ✗ | ✗ | ✗ | 90.00 ± 0.23 | 89.37 ± 0.19 |
| 2D FILM (Sec 2.1, Fig 2) | ✗ | ✗ | ✗ | 67.26 ± 0.78 | − |
| | ✗ | ✓ | ✗ | 79.69 ± 2.05 | 79.14 ± 2.35 |
| 3D FILM, projection (Sec 2.2.1, Fig 4) | ✓ | ✓ | ✗ | 65.49 ± 1.46 | 65.10 ± 1.67 |
| | ✓ | ✗ | ✓ | 86.92 ± 2.00 | 86.89 ± 2.04 |
| | ✓ | ✓ | ✓ | **89.98 ± 0.59** | **89.91 ± 0.73** |

## 5 CONCLUSIONS AND BROADER IMPACTS

We address the problem of answering a question from a single image, posed in a reference frame that is different to the one of the viewer. We illustrate the difficulties here in a controlled setting, proposing a new CLEVR dataset and exploring a 3D FILM-based architecture that operates directly on latent feature volumes (using camera conditioning via FILM or via direct rigid transforms on the volume). We propose two techniques to train volumetric encoders: with 2D-to-3D projection of ImageNet features, or using a self-supervised contrastive approach. In the latter case, we showed that the use of combined 2D+3D data augmentation was crucial to learning a volumetric encoder, without the use of camera extrinsics nor a postprocessor. Through rigorous ablations, we demonstrated that performing 3D FILM was the most effective for *CLEVR-MRT*, especially when the volume can be subjected to rigid transformations in order to answer the question.

Endowing intelligent embodied systems with the ability to answer questions regarding properties of a 3D visual scene with respect to the perspective of another agent could make such systems safer. In the case of an autonomous vehicles, better control decisions could eventually be made. If such systems are adversarial in nature, negative outcomes could arise to the adversaries of such systems.

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
