# OpenReview forum: "Visual Question Answering From Another Perspective: CLEVR Mental Rotation Tests"
_ICLR.cc/2021/Conference — Reject_

### Official Review · AnonReviewer1 · 2020-10-27
**The authors propose to learn mental rotations via a synthetic CLEVR-Mental Rotation dataset based on VQA**

**Rating:** 6
**Confidence:** 4

**Review:**

Pros:

1. The paper presents an interesting idea to learn mental rotations using a variation of the CLEVR-VQA dataset. The contributions are - the creation of this synthetic CLEVR-Mental Rotation dataset for targeting this problem and a model that encodes questions and viewpoint information to produce answers via FiLM based encoders and 3D volume encoder.
2. The results in Table 1 and Table 2 show improvements with respect to the baselines using their final model but there is still some concern in the improvements on their ablations.
3. The paper is well written and easy to understand.

Cons:
1. The motivation of why we need to learn mental rotations is not very clearly expressed, the practical examples given in the introduction are not sufficient. Does the model really learn these mental rotations from a simple spatial VQA task? This should be evaluated in the experiments either using activation maps or by visualizing intermediate 3D encodings.
2. Is the model trained on all views for a single question-view pair or any one random viewpoint is sampled during mini batch training ? Does the rotation of a scene done over the complete 360 degree ? How do you decide how much to rotate to generate a viewpoint ?
3. The self supervised learning of 3D volumes is an interesting idea, but it's use case in this particular problem is very weakly motivated both in experiments and theory. Why is this method better than the method discussed in Section 2.2.1? What is 3D data augmentation and how is it different from 2D data augmentation?
4. There is a large variance in some experiments in Table 1. Is it due to the camera transformation embedding? It will be good to discuss the reasons why this is in Table 1 and not in Table 2.
5. Although the models developed are used in a very different problem setting with minor contributions, still a large part of the methods seem to be derived from the literature.
6. The final results in Table 2 though argued are better due to small variance but more extensive experiments need to be performed to show the benefits of the self-supervised pre-training over the traditional encoder approach.
Minor: What is the value of t (tau) used in Eq 3 ? In Table 2 it shows 1.0, but in the text it’s discussed as 0.1. Is this a typo or both of them are supposed to be different, if yes why ?

---

> ### Author Response · Authors · 2020-11-17
> **Response**
>
> **The self supervised learning of 3D volumes is an interesting idea, but it's use case in this particular problem is very weakly motivated both in experiments and theory. Why is this method better than the method discussed in Section 2.2.1? What is 3D data augmentation and how is it different from 2D data augmentation?**
>
> The contrastive experiments is motivated by the fact that (1) we can learn from scratch a volumetric encoder from 2D images without performing any re-rendering tasks or assuming camera knowledge (n.b: while we leverage viewpoint camera in the actual FILM stage, it is not used at all during contrastive encoder pre-training); and (2) it addresses the issue where some of the experiments in Table 1 had slightly inflated variances, which we strongly conjecture is simply due to the fact that the domain mismatch between the pre-trained ImageNet encoder and CLEVR-MRT. Instead, here we pre-train an encoder on the same dataset that we train 3D FILM on, and the variances indicated in Table 2 support this conjecture.
>
> The differences between {2D,3D,2D+3D} augmentation is explained in Section 2.2.3. To re-iterate, we make a distinction between scenes and images: each scene comprises many images (views). For the sake of simplicity let us omit minibatches, and just consider individual examples. At each iteration of training, let S1 and S2 denote two random sampled scenes, and we sample x1, x2 ~ S1 (two views from S1) and y ~ S2 (one view from scene 2):
> - 2D data augmentation means: pull T(x1) and T(x1) close together**, push T(x1) and T(y) far apart. Note that x2 is never used here.
> - 3D data augmentation means: pull x1 and x2 close together, push x1 and y far apart. Note that here we do not use a stochastic augmentation function T -- the only stochasticity with respect to S1 is in sampling different views x1, x2 ~ S1
> - 2D + 3D: pull T(x1) and T(x2) close together, push T(x1) and T(y) far apart. This is just 3D data augmentation but with T() added back in.
>
> Training with 3D data augmentation essentially teaches the contrastive encoder how to distinguish between *scenes*, i.e. it can detect whether some random image pair (x,y) belongs to the same scene or not, which imbues such an encoder with strong 3D reasoning properties. You do not get this with 2D-only because the contrastive loss is never trained to pull the encodings of x1 and x2 together. However, we found that if you combine this with 2D data augmentation, the contrastive encoder learns sufficiently good features that it can be used in the FILM stage to achieve our best result (87%).
>
> To the best of our knowledge, the use of a volumetric contrastive encoder whose latent volumes can be subjected to rigid transformations in a downstream task (VQA) is novel.
>
> **This is not a typo, T() is a stochastic function so T applied to the same image twice (T(x1) and T(x1)) does not necessarily give two identical images!
>
> **"There is a large variance in some experiments in Table 1. Is it due to the camera transformation embedding? It will be good to discuss the reasons why this is in Table 1 and not in Table 2."**
>
> Hi, see point (1) in the official comment at the top of this page.

---

### Official Review · AnonReviewer2 · 2020-10-28
**The idea is not entirely new and details are missing**

**Rating:** 4
**Confidence:** 4

**Review:**

Summary:

The paper studies visual question answering focusing on answering questions in a reference image of a different viewpoint. They propose a new dataset CLEVR-MRT drawing motivation from the well-known visual reasoning dataset CLEVR to illustrate the idea in which they have full control of the changes of viewpoints in an image. They then propose to use a volumetric encoder to represent 3D image features of an image via either 2D-to-3D projection or a contrastive-based encoder and further adapt an existing method (FiLM) to handle 3D tensors. Experiments on the CLEVR-MRT show that the use of the 2D features and 3D features of an image is complementary to each other.

Comments (Technical, Major Flaws of this paper):

(1) The idea of addressing VQA in multi-view settings is reasonable but it is not entirely new. My main concern is at the limitations of a synthetic dataset in a controlled setting where the relations between objects are limited compared to real data. In addition, I believe that given enough such generated question-answer pairs with associated programs, models may possibly learn to decode the generation procedure under the hood instead of learning the actual semantic meanings of languages and the relations between objects.

(2) Since there are no statistics about the newly introduced dataset, it is hard to judge the empirical results in the paper. As pointed out by many previous studies (e.g. Hudson, D.A., et al., 2019; Le, T.M., et al., 2020), models' performance seems to converge on CLEVR given enough training data. Having that said, existing methods easily fail if we reduce the number of training instances. As for the CLEVR-MRT, even without any information about the viewpoints, the baseline models could achieve more than 70% accuracy on the proposed dataset. It seems that the dataset is too simple that the model could have good performance without knowing the camera parameters. This leads to concerns about the validity of the proposed dataset. Please address these points.
References:
 - Le, T. M., Le, V., Venkatesh, S., & Tran, T. (2020). Dynamic Language Binding in Relational Visual Reasoning. In IJCAI 2020.
 - Hudson, D. A., & Manning, C. D. (2018). Compositional attention networks for machine reasoning. In ICLR 2019.

(3) For those who are not familiar with the CLEVR dataset, briefly explaining the procedure to generate the dataset and its variants might be helpful.

(4) Given a question related to the object positions, there may exist many different views that provide the same answer. Let's take the question "How many green spheres to the left of the shiny gold thing?" in Figure 4 as an example. There are many views in the scene that provide the correct answer "1" for this question.   Without restricting the variance of the camera view (as in [1]), how can we ensure the model to infer the correct viewpoint?

Some typos:
- (1): 1. Introduction: We use the the Compositional -> We use the Compositional
- (2): 2.1 FILM Baseline: the viewpoint and canonical view is the same thing -> the viewpoint and the canonical view are the same thing
- (3): Figure 2: The dotted border on the ResNet-101 indicate -> The dotted border on the ResNet-101 indicates
- (4): Conclusion: In the case of an autonomous vehicles -> In the case of autonomous vehicles

---

> ### Author Response · Authors · 2020-11-17
> **Response**
>
> **"Since there are no statistics about the newly introduced dataset, it is hard to judge the empirical results in the paper."**
>
> Section X discusses how many scenes/questions are for each split, the range at which azimuths are sampled for the camera, as well as stating what question types in the original CLEVR dataset were filtered out.
>
> **"As for the CLEVR-MRT, even without any information about the viewpoints, the baseline models could achieve more than 70% accuracy on the proposed dataset. It seems that the dataset is too simple that the model could have good performance without knowing the camera parameters."**
>
> Thanks for addressing this concern. Please see points (3) and (4) in the official comments on top of the page.
>
> **There are many views in the scene that provide the correct answer "1" for this question. Without restricting the variance of the camera view (as in [1]), how can we ensure the model to infer the correct viewpoint?**
>
> Strictly speaking, the objective of the paper is not to infer the ‘correct’ viewpoint. This is not to say that the model doesn’t do this, but rather it is not trained explicitly to do so like in the case of re-rendering. Rather than train the model to accurately re-render a new viewpoint, we are simply asking it to perform a sequence of 3D reasoning steps (i.e. the camera-conditioned transform on h followed by 3D conv FILM blocks) such that it is able to answer the question correctly.
>
> We also address this in point (3) at the top of the page.

---

### Official Review · AnonReviewer4 · 2020-10-28

**Rating:** 4
**Confidence:** 3

**Review:**

The paper explores the problem of visual question answering from another perspective. Similar to VQA, a system is provided with a scene and a question. However, the difference is that the question needs to be answered from a viewpoint different from the one provided. Hence, the system needs to perform “mental rotation”. The paper creates a new dataset called CLEVR Mental Rotation Tests which is based on the prior CLEVR dataset. The paper also studies the efficacy of various supervised and self-supervised models on the proposed dataset.

#### Strong points:
- The problem of asking questions related to “mental rotation” seems interesting.
- The paper shows that contrastive pre-training could be useful for the task, which is an interesting result.

#### Weak points:
- Although the problem seems interesting, I am unclear about the usefulness of the proposed dataset. The paper says that “many computer vision systems could benefit from neural architectures that demonstrate good performance for more targeted mental rotation tasks.” To justify this claim it gives the following example, “given the camera viewpoint of a (blind) person crossing the road, can we infer if each of the drivers of the cars at an intersection can see this blind person crossing the street?”. This could potentially be a useful scenario, however, the dataset proposed is different from the example as the camera viewpoint is provided as part of the input and not inferred from the question. The paper does not provide justification or evidence of how the current setup (i.e. with camera viewpoint) is useful. In particular, it would be nice if the paper could further explain how the current setup is better than solving the “view rendering” and VQA problems separately.

- The dataset seems to be too simple for the mental rotation tests. It is unclear if in the future the dataset would be useful in distinguishing which models are better. As the paper shows that “2D baseline without camera conditioning” already achieves 70% accuracy. As far as I could understand, even without knowing which view to look at, a model could achieve 70% accuracy indicating that there is a lot of bias in the dataset. Moreover, simply adding camera embedding with the question to a 2D baseline (Table 1, 2D FILM with camera), already performs close to the best 3D model and upper bound. (Please clarify if my understanding is wrong.)

- The paper is poorly organized and hard to follow. For example, one of the contributions of the work is the CLEVR-MTR dataset, however, there is no clear section in the main paper describing the details of how the dataset. Instead, the information about the dataset is scattered in the introduction and related work. Another example is that the paper moves into talking about the method (Section 2) without defining the task concretely. It is only from the figure that one notices that the camera viewpoint is part of the input. From the examples provided in the introduction, the reader is under the impression that the camera viewpoint has to be inferred from the question itself. Similarly, it's hard to parse what the training signal for each baseline is. Does a baseline use the rendered image from the other view during training?

#### Minor Comments:
- The figures and tables are interspersed with the text making the paper harder to read. It might be better to place the figures and tables at the end of the beginning of the page so that the captions are separated from the main text.
- Many equations like some parts of equation 1 and equation 2 might not be necessary as they don’t seem to contribute to understanding the paper. In many places, it seems like a simple intuitive explanation would be sufficient.
- Similarly, Figure 3 might not be necessary.
- The figures are unclear and hard to understand. For example, is the canonical viewpoint part of the input? If not, Figure 2 and Figure 4 could be changed to make it more clear.
- Is this line correct, “If we add camera conditioning via FILM (that is, appending the camera embedding to the GRU’s embedding) then we achieve a much greater accuracy of 69.60 ± 0.09.” Should its value be 83.68 ± 1.21 as indicated in the table?

#### Overall Recommendation:
Although the problem could potentially be useful, the current dataset seems to be not so useful and over-simplified. Moreover, I found the paper not well-organized and hard to understand even after multiple reads. I feel the paper can be improved a lot and hence recommend rejection for the current version.

#### Post Rebuttal
(Copying from the discussion below)

I would like to thank the author(s) for their response. After going over them, I am still not very confident about the paper would stick to my initial assessment. Following are my primary concerns:

"We note that there is a distinction between wanting to see something from another point of view, versus wanting to answer a question from another point of view. The former is where re-rendering is appropriate, but we do not make the claim that this alternative (view rendering + VQA) performs better or worse empirically."

I understand the distinction. But the issue still remains. Why is the out-of-the-box "view rendering + VQA" solution insufficient? Is there any empirical justification for it? If not its hard to see the value in the current setup. A potential way to address this could be to run a simple out-of-the-box "view rendering + VQA" baseline.

"(2) R3 and R4’s concern about camera information being provided to the model and its potential infeasibility in practice: In real world settings, camera rigs can and do have knowledge about where they are situated in the world, for instance using SLAM or GPS coordinates. In that case, it is not unreasonable for e.g. an autonomous vehicle to answer queries by performing rotations and/or translations of its current viewpoint."

The concern was not about the viewpoint of the observer but the new viewpoint from which the question has to be answered. Also, the location of the new viewpoint need not be converted into float and appended to the question. It could be expressed in natural language. For example "viewpoint of the driver in the other car" like in the example provided by the paper. In the current setup the information about this viewpoint is provided in terms of exact coordinates, which makes the setup less interesting and not so practical.

Although the authors improved some of the figures, the latest version of the paper does not seem to address other clarity concerns like a clear section for the dataset; organization of text and figures; removing unnecessary equations

---

> ### Author Response · Authors · 2020-11-17
> **Response**
>
> **As far as I could understand, even without knowing which view to look at, a model could achieve 70% accuracy indicating that there is a lot of bias in the dataset. Moreover, simply adding camera embedding with the question to a 2D baseline (Table 1, 2D FILM with camera), already performs close to the best 3D model and upper bound. (Please clarify if my understanding is wrong.)**
>
> Hi, yes but there are some caveats to this that are worth knowing. See points (1) and (3) in the official comments at the top of the page.
>
> **Similarly, it's hard to parse what the training signal for each baseline is. Does a baseline use the rendered image from the other view during training?**
>
> We apologise for any confusion here. We will make this more clear as well as the figures. To explain:
>
> For the “upper bound” canonical view baseline, during all phases (train/valid/test), the dataset used is one where only the canonical view exists for each image. So there is no random sampling of views going on, and it is essentially equivalent to vanilla CLEVR (though not exactly because vanilla CLEVR has questions that are invariant to camera viewpoint such as counting, and we removed those).
>
> For all other experiments, during all phases (train/valid/test), each image in the minibatch is a randomly sampled view from a randomly sampled scene. Each scene contains 20 pre-generated camera views whose azimuths were sampled at random with a Uniform(-180, 180) distribution. Note that the canonical view (at azimuth=0) is an extra view (so there are actually 21 views), but for all experiments apart from the “upper bound canonical” one the training procedure pretends the canonical view does not exist (so we can pretend each scene has 20 views, not 21). However, because the 20 pre-generated camera views were sampled from a Uniform(-180, 180) it *can* be the case that by coincidence the network is given a view that is close enough to canonical in the sense that answering the question is relatively straightforward. To answer your question, only one input image is fed through the network and that is the viewpoint camera (not the canonical).
>
> For each FILM experiment, the only inputs are:
> - The viewpoint image
> - The viewpoint camera. This is just a camera matrix wrt to world coordinates, with 6 values (3 denoting pose on x/y/z and 3 denoting translation on x/y/z).
> - The question, posed with respect to the canonical viewpoint
>
> Therefore, the difference between experiments is really what the network does with the camera coordinates of the viewpoint camera In Table 1, we illustrate what the supervisory signal is with these columns:
> - “camera (embed)” means that we feed this camera through a trainable MLP to produce a camera embedding that is subsequently passed to the FILM blocks.
> - “camera (rotation)” means that we feed this camera through a trainable MLP which produces another camera matrix describing the *relative transform* between the current viewpoint and the canonical, and this is used to rotate/translate the feature volume before it is passed to FILM.
>
> **This could potentially be a useful scenario, however, the dataset proposed is different from the example as the camera viewpoint is provided as part of the input and not inferred from the question. The paper does not provide justification or evidence of how the current setup (i.e. with camera viewpoint) is useful.**
>
> We address this concern (in part) in point (3) in the official comments.
>
> Indeed, it would be possible to run experiments on a new version of the dataset where the canonical viewpoint is described in the question, e.g. “How many red cubes are there to the left of the green sphere *when I rotate my viewpoint by X degrees and translate by Y units?”, however this is just converting floats in a camera matrix to plain language and appending it to the question string. Converting floats to strings may be problematic however because that probably will not generalise well (i.e. does the RNN know the relationship between the floats represented as strings “1.54” and the string “1.56”? It would have to learn how to do arithmetic). The alternative is to separate the camera coordinates from the question, which is precisely what we are doing now. Furthermore, to reiterate point (3) in or official comment, it is not unreasonable for a camera rig to know where it is oriented in the world, and simply use the coordinates directly to make some sort of inference.

---

> ### Author Response · Authors · 2020-11-17
> **Response**
>
> **In particular, it would be nice if the paper could further explain how the current setup is better than solving the “view rendering” and VQA problems separately.**
>
> Thank you for raising this point, we should clarify this in the paper. What we try to illustrate in our paper is that one can perform this style of VQA without ever having to explicitly perform a re-rendering via the training of a decoder network. For example, for either of the encoder networks proposed (the ImageNet encoder or the contrastive), it would be possible to train a decoder using reconstruction loss on ground truth images, e.g. in addition to the FILM model which conditions on h (after the postprocessor), another branch conditions on it to perform decoding/re-rendering. Not only is this much more computationally expensive however, it may even be to the detriment to the FILM model since now we have to carefully strike a balance between good VQA performance and good reconstruction.
>
> We note that there is a distinction between wanting to see something from another point of view, versus wanting to answer a question from another point of view. The former is where re-rendering is appropriate, but we do not make the claim that this alternative (view rendering + VQA) performs better or worse empirically.
>
> **"...however, there is no clear section in the main paper describing the details of how the dataset"**
>
> Details on how the dataset was generated are described at the end of Section 3 (though perhaps for clarity should be moved into its own subsection!).
>
> **Concerns about the clarity of writing and organisation:** thank you, we will address these concerns.

---

### Official Review · AnonReviewer3 · 2020-10-28

**Rating:** 4
**Confidence:** 3

**Review:**


### Overall

Authors extend CLEVR dataset so as to consider multiple viewpoints, and evaluate current neural network models in that setting. They also update a standard approach to introduce camera viewpoint information in the network so it can better answer visual question from the canonical scene frame even from other perspectives.

### Positive aspects

* Authors provide a study on a important topic of Computer Vision: understanding multiple views of a same scene. They do such study on a hard task, which is VQA. Actually, authors provide a more complex version of a simple VQA dataset (*simple* because it is synthetic and has very well established domain limits).
* Authors evaluate different training frameworks (supervised and unsupervised from scratch).
* Authors provided accuracy values for pretraining with NCE, which can be helpful.
* Results seem to be promising.
* In general, text is well written and easy to read.
* It is interesting that even a frozen pretrained network provides good results in such visually different dataset. Although, it was nice that authors trained an encoder from scratch.
* Code already available!

### Weak aspects and suggestions

* The problem is interesting, though my main concern is regarding the novelty and contribution of the paper. It seems to be an adaptation of CLEVR dataset, and an adaptation of the FILM model. In addition, authors use camera viewpoint information to ease the identification of the scenes. I have mixed feelings in using such specific kind of information in the model, because in a real world scenario we don't have access to them. I might be wrong, but maybe it is possible to insert a module in their approach to estimate the camera parameters, so as the network itself could learn to predict how viewpoints work and how scenes change with that. I think this could be done by adding such parameters as target information some of the models. For instance, the unsupervised architecture could be trained to predict whether the scenes are the same, but also the camera parameters. Apologies if I miss something here.

* The proposed architecture seems to be basically an adaptation of the FILM model considering camera viewpoint information.

* FILM (2018) is the best performing approach in CLEVER to date? There are more recent approaches that could be used in the results section as baselines.

* It is unclear what happened to the spatial-related questions. They were removed of the dataset?

* Results are promising, although why do they have such high variance? (7-8% of variance is not negligible by any means); considering that for some experiments it is likely that 2D FILM provides similar performance than 3D one. A statistical test might help to verify whether such results are statistically significant or not.

* Font size for all images should be quite larger. It is hard to read in the current size.

* Figure of the post processor does not help much. Authors could detail a little bit more what is inside that $postproc_w$ box.

* *"Since the post-processor is a learnable module through which the FILM part of the pipeline is able to backpropagate through, it can be seen as learning an appropriate set of transforms that construct 3D feature volumes h0."* I suggest rewriting this sentence, it is very confusing.

* *"While we obtained great results, it may not leave a lot of room to improve on top of our methods,"* This sentence is odd. The sentence "we obtained great results" can be written in a more objective and scientific way (avoid the usage of adjectives). Another important aspect is: often it is easy to provide first large steps in a task (ImageNet for instance), although it gets much harder to improve on that when results are good (AlexNet vs ResNet, see the performance difference). Another aspect: maybe authors made the task too easy and should have explored more challenging scenarios.

* *"and we identified some ways in which the dataset could be made more difficult"* Those ideas to make the task more challenging are indeed important. Why authors did not perform experiments in such scenarios? It does not seem very hard to generate such datasets.

* Is it possible to visualize and understand what the postproc module does? It would be nice to visually explain the $h'$ (64, 16, 14, 14) tensor represents.

* There could be some qualitative analysis.

* The dataset extension seems to be a large portion of the work. I think it could have a separate section with more details.

### Additional questions

* What happens if other conditioning camera information strategy is used? For instance, simply concatenating or using other simpler fusion techniques. FILM would perform much better than other simpler approaches?

* *"ResNet outputs... feature maps h of dimensions (1024,14, 14)"* Is this correct? I believe Resnet101 outputs (2048, 14, 14) feature maps.

* *"in practice, we found $\tau = 0.1$ to produce the lowest softmax loss."* Which ones you have tested? Why $\tau$ is 1.0 in Table 2?

* *"Another idea is to allow the viewpoint camera’s elevation to change. "* That is true. Or even the distance from the camera. Why did authors decide not to include such examples in this work?

* *"This is to be expected, considering that any camera information that is forward-propagated will contribute gradients back to the postprocessing parameters in the backward propagation, effectively giving the postprocessor supervision in the form of camera extrinsics."*. Can authors support/prove this claim?

---

> ### Author Response · Authors · 2020-11-17
> **Response**
>
> **It is unclear what happened to the spatial-related questions. They were removed of the dataset?**
>
> Do you mean the non-spatial questions? This is at the end of Section 3: “To focus on questions with viewpoint dependent answers, we filtered the set of questions to only include those containing spatial relationships (e.g. ‘is X to the right of Y’).” In other words, questions whose answers would be *invariant* to the viewpoint camera (e.g. how many green cubes are there) are not used in the dataset generation process.
>
> **Results are promising, although why do they have such high variance? (7-8% of variance is not negligible by any means); considering that for some experiments it is likely that 2D FILM provides similar performance than 3D one. A statistical test might help to verify whether such results are statistically significant or not.**
>
> Thank you for addressing this. See the point (1) in the official comment at the top of this page.
>
> **I have mixed feelings in using such specific kind of [camera] information in the model, because in a real world scenario we don't have access to them.**
>
> See point (2) in the official comment at top of page.
>
> **Another aspect: maybe authors made the task too easy and should have explored more challenging scenarios. Due to time constraints we were unable to re-run all experiments on any new version of the dataset prior to the submission deadline. However, we have now generated a slightly more complex version, see official comment.**
>
> See point (4).
>
> **ResNet outputs... feature maps h of dimensions (1024,14, 14)" Is this correct? I believe Resnet101 outputs (2048, 14, 14) feature maps.**
>
> Thanks! We should have clarified that this is the ResNet-101 with the last ‘block’ chopped off (the PyTorch version is split into four ‘blocks’ each with 3, 4, 23, and 3 modules inside, respectively, so we remove those last 3 modules). Therefore this gives 1024 feature maps rather than 2048.
>
> **Is it possible to visualize and understand what the postproc module does? It would be nice to visually explain the h′ (64, 16, 14, 14) tensor represents.**
>
> Sure.
>
> x -> [frozen resnet encoder] -> [postprocessor] -> [rotation] -> [FILM blocks]
>
> For our “3D FILM + projection” architecture (Sec 2.2.1, Fig 4), a postprocessor is needed since we are piggybacking on top of an encoder that was pretrained on ImageNet (an image/2D dataset). Using PyTorch-like shape notation and omitting the batch axis, the output of the ResNet encoder, for an input of (3,224,224), is (1024, 14, 14), i.e. 1024 feature maps of spatial dimension 14x14. The first thing that happens is that this tensor gets ‘projected’ into 4D via a reshape operation, so (1024//16, 16, 14, 14) = (64, 16, 14, 14). In other words, we now have 64 feature ‘cubes’ of size 16x14x14. This then goes through multiple 3D conv blocks (conv3d-BN-relu) to produce the output feature cube of the same dimension (64,16,14,14).
>
> **This is to be expected, considering that any camera information that is forward-propagated will contribute gradients back to the postprocessing parameters in the backward propagation, effectively giving the postprocessor supervision in the form of camera extrinsics.". Can authors support/prove this claim?**
>
> We ran an extra ablation on our best “3D FILM, projection” experiment, which is the number you see in parentheses (with the † symbol) underneath the bolded result in Table 1. In this ablation the postprocessor was left randomly initialised, i.e. its parameters were not updated during training. This means the postprocessor simply performs a random projection. This achieved 68.98% accuracy on the validation set. Reiterating our architecture for “FILM + projection”:
>
> x -> [frozen resnet encoder] -> [postprocessor] -> [rotation] -> [FILM blocks] -> prediction
>
> Whether we are using the camera coordinates to condition the [rotation] op, or passing the camera coordinates to the FILM blocks (not shown here, but see Fig 4),  camera information is being used in the forward pass of the network, and that subsequently influences the gradients that are back-propagated. Because the postprocessor is not frozen, it also receives those gradients and therefore are updated based on that information.
>
> **in practice, we found τ=0.1 to produce the lowest softmax loss." Which ones you have tested? Why τ is 1.0 in Table 2?**
>
> Thanks for spotting this error, indeed it should be 0.1.

---

### Author Response · Authors · 2020-11-17
**Key concerns addressed by reviewers**

Hi,

We thank all reviewers for their highly detailed feedback and comments. We appreciate that reviewers found that the mental rotation VQA task was interesting and/or important (R3, R4, R1); that the contrastive pre-training was an interesting contribution (R3, R4); that the results are promising (R3, R4, R1); and that the text was clear (R1, R3). In addition to responding to specific reviewers about concerns, there were some key concerns shared by several reviewers, so we will address those here.

**(1) Reviewers (R3, R1) noted that some of the results in Table 1 (namely the 3D FILM ones) had unusually high variance, bringing up concerns about statistical significance of the proposed methods.**

We should have noted in the original submission that the inflated variance is due to few runs running into local minima and plateauing at a relatively low accuracy. We note that the inflated variance only appears to be the case for when we use the ImageNet encoder with 3D feature transformations, and does not appear to happen for our results for when we use the contrastive pre-training procedure. It is highly likely that this is due to a domain ‘mismatch’ between ImageNet and CLEVR, because the ImageNet encoder remains frozen during training.

For example, if we take our best performing method in Table 1 (FILM + projection, using camera for rotation), find the best experiment with hyperparameter tuning and re-run that same experiment over 5 more seeds (for a total of 6 runs), the max validation set accuracy obtained by each is: **[0.93, 0.90, 0.93, 0.93, 0.90, 0.70]**. The mean/stdev of these numbers is 88.13 +/- 8.01, as reported in the original submission pdf. However, the variance has been severely inflated by the last run whose highest accuracy was 70%. In retrospect, we should have highlighted this phenomena in the discussion section, as well as account for these outliers when computing the mean and variance. Therefore, we have updated those results by computing the mean/stdev over the best 3 performing models on the validation set. We have updated the numbers in Table 1 to reflect this, which is reflected in the revised pdf.

**(2) R3 and R4’s concern about camera information being provided to the model and its potential infeasibility in practice:**

In real world settings, camera rigs can and do have knowledge about where they are situated in the world, for instance using SLAM or GPS coordinates. In that case, it is not unreasonable for e.g. an autonomous vehicle to answer queries by performing rotations and/or translations of its current viewpoint.

**(3) (R4, R3, R2) Concerns about the simplicity of the dataset, primarily motivated by the 2D no-camera baseline which achieves 70%, but also concerns about strong biases in the dataset.**

We appreciate that this concern was mentioned. It is not uncommon for VQA datasets to contain strong biases and this phenomena is not limited to CLEVR or its derivative datasets. It is an active field of research and is most commonly seen in the literature as language-based biases (see [1,2,3,4]) (i.e. the situation where VQA models may exploit statistical regularities in the question rather than considering what is in the image), although the source of bias is certainly not limited to just language. We argue that whether or not the no-camera baseline accuracy of 70% is problematic is highly dependent on the problem that we are attempting to solve. For example, in safety-critical applications such as autonomous vehicles or blind person navigation, 30% error rate is high enough to be very problematic in a production setting. Because removing sources of bias in a dataset can be virtually impractical (especially if it is obtained in a real world setting), we decided to simply make these biases known in our results, which is why we included in Table 1 the majority class baseline and trained a question-only (RNN-only) baseline. Lastly, after we excluded outlier experiments and computed the mean/stdev accuracy over the top 3 models (see point #1 in this comment ), our best result achieved 92.80 +/- 0.30. This is a rather considerable difference to 70%.

- [4] Agrawal, A., Batra, D., Parikh, D., & Kembhavi, A. (2018). Don't just assume; look and answer: Overcoming priors for visual question answering. In Proceedings of the IEEE Conference on Computer Vision and Pattern Recognition (pp. 4971-4980).
- [1] Ramakrishnan, S., Agrawal, A., & Lee, S. (2018). Overcoming language priors in visual question answering with adversarial regularization. In Advances in Neural Information Processing Systems (pp. 1541-1551).
- [2] KV, G., & Mittal, A. (2020). Reducing Language Biases in Visual Question Answering with Visually-Grounded Question Encoder. arXiv preprint arXiv:2007.06198.
- [3] Manjunatha, V., Saini, N., & Davis, L. S. (2019). Explicit bias discovery in visual question answering models. In Proceedings of the IEEE Conference on Computer Vision and Pattern Recognition (pp. 9562-9571).

---

### Author Response · Authors · 2020-11-17
**Results on a version of CLEVR-MRT with more variability**

**(4) Concerns about the simplicity of the dataset (R4, R3, R2), part II:**

We generated a modified version of CLEVR-MRT that exhibits more variability than in the original dataset in order to further probe our proposed models and baselines. In the original dataset the only source of variability in the camera was in its azimuth, i.e. its orbit around the scene. However, this was at a fixed elevation of 30 degrees. In the modified version of the dataset, we also made the camera elevation stochastic (N.B: the canonical camera stays at the same elevation, and is unaffected), so that it is sampled from a Uniform(20,30). This means that in some cases the camera will be lower than it normally is. We also enabled small objects (the original dataset had only large-sized objects). Both of these  aforementioned additions can potentially increase occlusion effects (i.e. making it difficult to answer questions), but it is a better reflection of a real-world setting. **While we have uploaded a new draft and posted new numbers in Table 3 as well as an example scene of the modified dataset in Figure 6, we want to stress that hyperparameter tuning for some of these experiments are still ongoing.**

Comparing Tables 1 and 3, whereas the 2D FILM baselines for {no camera, camera} were ~70% and ~84% respectively, they are now (in Table 3) ~67% and ~80%, respectively.

---

### Decision · Program_Chairs · 2021-01-07
**Final Decision**

**Decision:**

Reject

**Comment:**

This paper was reviewed by 4 experts in the field. The reviewers raised their concerns on lack of novelty, unconvincing experiment, and the presentation of this paper, While the paper clearly has merit, the decision is not to recommend acceptance. The authors are encouraged to consider the reviewers' comments when revising the paper for submission elsewhere.